# Vital Conjunctures in Compound Crises: Conceptualising Young People's Education Trajectories in Protracted Displacement in Jordan and Lebanon

**Zoë Jordan** *,†  and **Cathrine Brun** †

Centre for Development and Emergency Practice, Oxford Brookes University, Oxford OX3 0BP, UK; cbrun@brookes.ac.uk
* Correspondence: zjordan@brookes.ac.uk
† The two authors contributed equally to this work.

**Abstract:** This paper emerges out of a study with 293 young people (Syrians, Palestinians and nationals) living in contexts of compound crises and protracted displacement in Jordan and Lebanon. In the paper, we discuss how young people's education trajectories can be conceptualised, operationalised and studied. We synthesise different approaches to understanding and analysing such trajectories into a framework that captures the intricate and multi-directional ways that young people navigate towards uncertain futures. The framework on multi-directional trajectories takes its starting point from an understanding of Victoria Browne's 'lived time', captured through how different temporalities come together in one person's story. After presenting our framework and the context in the first part of the paper, the second part applies the framework to analyse the 'vital conjuncture' of leaving education. By analysing leaving education as lived time, we create nuanced insights into how this vital conjuncture can be understood to shape young peoples' trajectories. In conclusion, we discuss the value of understanding trajectories as lived time by illuminating how young people experience and navigate their education trajectories.

**Keywords:** trajectories; youth; lived time; education; vital conjunctures; Jordan; Lebanon; displacement

## 1. Introduction

There is wide consensus on the need to turn away from prescriptive models of youth based on age or linear stages towards a more complex notion of youth transitions (Punch 2002; Valentine 2003; Hörschelmann and Schäfer 2005; Jeffrey 2010). Moving from education to employment is one of the most important transitions from childhood to adulthood, alongside household and family formation (Jones 2000 in Devadason 2007; Pallas 2003). Young people's transitions through education are diverse, complex and messy, and there is still a need to find ways of capturing these multi-directional transitions. It is this challenge we are responding to here: to establish a way of analysing young people's multidimensional and multidirectional education trajectories. By focusing particularly on the multifaceted, stretched-out process of leaving education for young people living in displacement, the paper aims to understand some of the vital processes that shape young people's trajectories.

'Trajectory' can be understood geographically as a particular path taken through an environment, implying both position and momentum. However, it can also be understood through a temporal lens. Our starting point is to approach trajectories as tools to capture the different transitions in a life history and the rich set of experiences, practices and navigation that takes place within and between those transitions, thereby shaping their nature. To achieve this, we develop a conceptual model for analysing trajectories based on young people's 'navigation' (Vigh 2009) of 'vital conjunctures' (Johnson-Hanks 2002), understood through the multiple lenses of 'lived time' (Browne 2014). Leaving school is

applied as an example of how vital conjunctures emerge and influence the multidirectional trajectories of young people's lives.

We contribute to scholarship that seeks to understand how young people—as a broad social and cultural category—live in uncertain contexts of displacement and crisis, experience and move through their childhood into youth and navigate their lives towards potential futures (Jeffrey 2009; Bellino 2018; Sommers 2012).

In contexts of crisis, the 'normal' frameworks and rhythms of time may become problematic, disrupted or transformed. Understanding experiences of temporality in such circumstances can help us to understand these experiences and the dynamics of crisis (Cwerner 2001). We therefore also contribute to the growing body of work on time and temporality in forced migration, and particularly to the emphasis on how temporality shapes the experience and practices of displacement (Griffiths 2014; Brun 2016; Fontanari 2017; Kallio et al. 2020; Poole and Riggan 2020).

'Youth' is a contested category with multiple meanings, and age is an insufficient approach to understanding the meaning of youth (Furlong 2016; Hart 2014). Yet, in the material on which this paper is based, we deploy an age category of young people commensurate with the official definition of youth in Lebanon and Jordan: 15 to 29 years old. The material for this research was collected as part of a research programme titled "From Education to Employment? Trajectories for young people in Lebanon and Jordan in the context of protracted displacement". The research team is situated across Lebanon, Jordan and the United Kingdom and we conducted 293 interviews with young people in Jordan and Lebanon—Jordanian and Lebanese nationals as well as Syrian and Palestinian refugees in both countries. We discuss our methodological approach to understanding young people's trajectories further below.

Lebanon and Jordan are both protracted displacement contexts. They host some of the largest numbers of refugees from Syria per capita (1 in 7 in Lebanon, 1 in 15 in Jordan (UNHCR 2019)), and both have large Palestinian and Palestinian-origin populations. In Jordan 2,206,736 Palestinians are registered as refugees with UNRWA (UNRWA 2021a, 2021b), while in Lebanon estimated numbers range from 174,422 to 280,000 Palestinians (Chaaban et al. 2016; LDPC 2018). People in both countries have lived through repeated and compound crises resulting from displacement, economic shocks and political uncertainty. In October 2019, people in Lebanon demanded an end to political and economic uncertainty, turmoil and inequality with protests across the country. It was in the shadow of the uprising known as the Revolution in Lebanon that we interviewed young people about their trajectories from education to employment. In Jordan, there have been a number of protests in recent years, including political protests in 2011, Income Tax law protests in late 2018 and the 2020 Teachers' union protests. Since our interviews were conducted, the displacement and economic crises in both countries have been compounded by the COVID-19 pandemic, that has deepened structural inequalities, with further disruption to many young people's education. For young refugees and those growing up in the radical uncertainty of crisis and protracted displacement contexts, the connection between education and employment continues to be restricted by political, legal and social realities including limitations on access to education and the labour market (Herrera 2006).

In the following sections, we explain the education context for young refugees in Lebanon and Jordan, before drawing on the concepts of 'vital conjunctures', 'lived time' and 'navigation' to develop a framework for studying youth education trajectories in contexts of uncertainty and protracted displacement. Related to our notion of lived time, we introduce our methodological approach to constructing trajectories. In the empirical main body of the paper, we demonstrate the contribution of our framework to analysing young people's educational trajectories through the discussion of overall findings from the interviews with 293 young people (refugees and non-refugees) before zooming in on two young refugees' navigation of the critical conjuncture of leaving education. In conclusion, we reflect on the usefulness of defining trajectories as lived time and the nuances that analysing leaving school as a vital conjuncture contributes to the study of youth transitions.

## 2. Education in Contexts of Protracted Displacement and Crisis in Jordan and Lebanon

The international pressure for all children to participate in education and the success of these efforts have led to education becoming one of the defining features of modern childhood (Crivello 2009). At a global level, progress has been made on expanding access to primary education, but the enrolment of young people in secondary and tertiary education remains low (UNICEF 2019). For refugees, these figures are stark: only 24% of secondary-school aged refugees are enrolled, and just 3% access higher education (UNHCR 2019). Education in emergencies is therefore a growing area of interest and funding (Burde et al. 2017), yet much education for refugees is more about access than considerations of the quality and function of education (Brun and Shuayb 2020).

Lebanon and Jordan have both adopted policies to expand the provision of education for refugees and include them in existing school structures such as Reaching All Children with Education (RACE I and II) in Lebanon (MEHE 2014, 2016) and iterations of the Jordan Response Plan (JRP) (MoPIC 2015, 2016, 2017, 2020). Yet, the education systems in Jordan and Lebanon remain stratified according to legal status and socio-economic status. The enrolment of Syrian refugees in school in Lebanon and Jordan—particularly in secondary school and higher education—remains low (Al-Hroub 2011; Buckner et al. 2018). Syrian refugees in both countries can generally access primary education in public schools but mainly in afternoon shifts, often with lower quality and diversity of lessons than the morning shift. In Jordan and Lebanon, education for Palestinian refugees is separate from that of nationals and Syrian youth and is provided by the United Nations Relief and Works Agency for Palestine Refugees in the Near East (UNRWA), though not all Palestinian refugees attend UNRWA schools. In Jordan, UNRWA-provided schools are only available until the end of primary education, whereas in Lebanon there are a limited number of secondary schools and two vocational institutes. In addition to the formal sector, in both countries there are non-formal schooling programmes. These programmes support students who have dropped out of school but wish to continue their education, including catch-up initiatives and Home Schooling. In addition, there are a number of informal programmes offered by a range of actors that provide alternative learning and life-skills. It is unclear how many children are attending informal and non-formal learning programmes (Karasapan and Shah 2018). While formal vocational education has previously been largely overlooked, resulting in it being perceived as low-quality and undesirable by young people, it has been subject to reform in recent years in order to equip young people with work-oriented skills and is now established in both countries (MEHE 2018; ETF 2006; MoE 2017; UNESCO-UNEVOC 2019). However, access to vocational training for Syrian and Palestinian refugees is still restricted to private institutions and UNRWA institutions in Lebanon or short courses. While there are few restrictions on the specialisations available to refugees, the limited occupations open to them for employment constrain their training choices (MEHE 2018).

There are many challenges to staying in education, but, correspondingly, there are also challenges for those who leave or complete education. Structural barriers remain, preventing young people from entering the labour market and making for uncertain futures. There is rising youth unemployment globally, and Lebanon and Jordan are no exception (LCPS 2019; UNICEF 2019; UN 2020). For young refugees in Jordan and Lebanon, social and economic barriers to employment are formalised through legal restrictions on working. Non-nationals in Lebanon are required to have a work-permit to access employment. Under the Jordan Compact, the Jordanian government opened work permits for Syrians in 2016, allowing for employment within some specific sectors. However, many Syrian refugees have not applied for permits, and a large percentage are working informally. People of Palestinian origin in Jordan hold different statuses depending on when they arrived in Jordan and from where they moved.

In both Lebanon and Jordan, the substantial restrictions on staying in school and accessing employment affect young people's trajectories and influence the paths taken and how they plan their future. In the sections to follow, we set out to understand how we

can analyse young people's trajectories and the zones of possibility that emerge at specific points in those trajectories in the context of displacement and crises.

### 3. Plotting Trajectories in 'Lived Time': A Polytemporal Discussion of Vital Conjunctures and Navigation

Young people's trajectories move at different speeds and in different directions at different times without necessarily taking a coherent path to a predetermined end. 'Trajectory' may be an ambiguous concept, and a life story is often inadequately represented as an accumulated set of events that explain where an individual is today. Consequently, in this section, we set out to consider how lived time—reflecting a more multifaceted perspective on a life history—can be used to understand the vital conjunctures that plot a young person's trajectory.

Vital conjunctures are zones of possibility that emerge at specific periods in an individual's life where there is the potential for change or transformation (Johnson-Hanks 2002). As opposed to set life stages in the transition from adolescence to adulthood, vital conjunctures are key moments—migration, education choices, career change—in which the future is substantially at stake, and not necessarily clear or fixed. Deriving from major societal events in a person's life as well as from the smaller-scale, every day and more mundane experiences, conjunctures take place continuously and help to direct and redirect an individual's trajectory. Vital conjunctures are the major life events that the individual or society identify as significantly shaping the ways in which young people plot their trajectories (Jeffrey 2010; Johnson-Hanks 2002). The notion of 'vital conjunctures' opens up the possibility of capturing the multiplicity and complex temporality of trajectories as lived time and the different experiences of societally defined vital conjunctures. Different experiences, representations and practices of time come together in a trajectory. A 'polytemporal' understanding of lived time requires us to pay attention to how multiple times come together in the vital conjunctures that plot a trajectory. Inspired by (Browne 2014), we suggest that trajectories—as lived time—can be analysed by unpacking the different times that constitute them: narrative time, calendar time, the time of trace and generational time.

'Narrative time' focuses on the ways in which trajectories are temporalised through narrative configurations. Narrative time is 'the time of beginnings, middles and ends; flashbacks and flash-forwards; turning points and returns' (Browne 2014, p. 73). In narrating their experiences, young people contextualise and interpret their trajectories in relation to their present situation.

While narrative time suggests the experience and contextualisation of lived time, the temporalisation of history often takes place through 'calendar time', which 'organises histories into chronologies and timelines through temporal markers such as days, months, years, decades, and centuries' (Browne 2014, p. 99). It makes available a way of organising narratives through a common and shared temporal reference. Calendar time is not a neutral time indicator, but takes on a distinct meaning in any society and can be used as a political tool to organise the society and structure young people's lives (Cohen 2018). The modern education system epitomises calendar time: it structures and disciplines lived time for young people and helps to create particular hopes and aspirations for the future (Poole and Riggan 2020). Calendar time represents a 'power geometry of time' (Sharma 2014), where people in different positions have different control over their present and future and experience time differently (Griffiths et al. 2013).

The past affects our trajectories through documents, artefacts and events. The 'time of the trace' refers to a nonlinear, two-way temporality in which these past artefacts and events 'spill over' into, and give meaning to, the present (Browne 2014). Traces are one representation of the past in a young person's trajectory, but capturing a wider sense of continuity and change—beyond the individual trajectory—is fundamental to understanding lived time.

The final kind of time we address here, 'generational time', is relational and refers to how we can approach continuity and change between 'people of different ages and eras' (Browne 2014, p. 119). Generational time is about how memories are passed on

from one generation to the next, such as displacement histories, identities and allegiances. Generational time can therefore be used to define what binds generations together, how the past influences the present, the continuities and discontinuities between generations as well the diversity within a generation: generational time emphasises young people as the embodiment of time (Murphy 2012; Hart 2014).

Our final key concept is 'navigation'. By making lived time operational through these four times, we develop a format for analysing young people's trajectories by paying attention to the ways in which time is made and experienced and to how present realities and expected futures are navigated. The general uncertainty in young people's lives globally, and particularly for those who have been displaced or who are marginalised, means that trajectories do not always occur according to well-considered plans or 'ordinary' expectations (Kallio et al. 2020; Poole and Riggan 2020). In the context of protracted displacement in Jordan and Lebanon, characterised by uncertainty and changing regulations, the interaction between structures and agency evident in vital conjunctures is further pronounced, and such moments are fractured and extended. When studying how young people act within their environment, we also need to take into consideration the movement and changes within these environments (Vigh 2009). Focusing on the interactive movement of people and their environments is not to suggest that everything is adrift, but it helps to explain how people interpret and act within uncertain contexts to alter their circumstances and positions, to remove themselves from constraining conditions and to move towards preferable situations. Navigation encompasses agency, changing circumstances and accumulated knowledge. As such, it can be used to understand how young people move along, bring together, plan, anticipate and imagine their trajectories, through their actions at vital conjunctures.

## 4. Methodology: Capturing Lived Times

Understanding trajectories as lived time requires a methodological approach that can capture the vital conjunctures as key moments across different times. As set out above, narrative time is the interpretation of events and experiences seen in other forms of time (trace, generational and calendar). We utilised narrative time as an approach and tool, using a narrative interview as an entry point to capture young people's trajectories by situating those trajectories in the interpretation of the past and the future in the present understanding of the research participants.

The project started with a quantitative survey with 1400 young Syrian, Palestinian, Jordanian and Lebanese respondents. From here, the research team conducted 293 semi-structured life history interviews with young Syrians, Palestinians and nationals in Jordan and Lebanon (Table 1).[1,2]

**Table 1.** Research participants and life history interviews, according to country and legal status.

| | Jordan (145 Total) | | Lebanon (148 Total) | | |
| --- | --- | --- | --- | --- | --- |
| | **Male** | **Female** | **Male** | **Female** | **Total, by Nationality** |
| Jordanian or Lebanese National | 24 | 29 | 20 | 28 | 101 |
| Syrian | 25 | 25 | 22 | 16 | 88 |
| Palestinian, including Palestinians from Syria | 25 | 17 | 33 | 29 | 104 |
| Total, by gender and country of residence | 74 | 71 | 75 | 73 | 293 |

Situating their education and employment histories in the context of how the young people narrated and made sense of their life histories enabled us to map out vital conjunctures in relation to young people's education and employment trajectories. In analysing the interviews, we realised the complexity of their trajectories. We sought to capture the

movements through lived times and identified the stretching out of vital conjunctures over time, as we explain below.

In order to capture a wide range of trajectories, young people were recruited from rural, urban and industrial areas in the targeted locations (Bekaa, Beirut and Saida in Lebanon, Amman Governorate in Jordan), including Palestinian camps and gatherings and Syrian informal camps. Interview participants were recruited through snowball and convenience sampling and in accordance with a quota of nationality, gender and place of residence. The research team reflected a diverse mix of genders, nationalities, current places of residence, age and marital status, among other identities, as well as a range of education and employment histories. Where permitted, interviews were recorded and transcripts produced in English. The narratives are loosely framed through calendar time such as school grades and age in years, and take the form of a life history, focusing particularly on the period from entering education to the present (some are still in education, some work, some are in between education and work or following a different life path). Within this framework, we identify key events in young people's histories and context that they perceive as having influenced their trajectories. Following Browne (2014) and her reading of Said (1993), we analyse narratives contrapuntally, recognising the multiplicity of perspectives in one narrative in order to tease out the relational, political and temporal dynamics that underpin a particular narrative, keeping in mind that a narrative is always open for interpretation and always in the making.

## 5. Narratives of Leaving School

Leaving school is not a straightforward event that takes place as one decisive moment. From our survey data, we found that nearly 40% of respondents left education before obtaining a certificate that qualifies them for work (Shuayb et al. 2021). We also found that more refugees than nationals were leaving school. In Lebanon, males are 9% more likely to drop out than females, with no difference being reported in Jordan. In Jordan, basic education continues until Grade 10, when students are typically aged 16. In Lebanon, education continues until Grade 9 (Brevet), when students are 15. From our 293 qualitative interviews, there were 63 young people (21 in Jordan, 42 in Lebanon) who left education before completing compulsory schooling and who were above the typical age for their countries' education system. Again, refugees, and particularly Syrian refugees, are over-represented in the groups leaving school early.

Much of the literature on dropping out of school concentrates on explaining the demographic and socio-economic determinants of dropouts with the view to keeping youth in school longer and avoiding the negative effects on the individual, family and society of dropping out (Hong 2020; Johansson 2019). The narratives of our participants confirm the salience of demographic and socio-economic determinants to a certain extent, although the timing, the experience and the reasons and degree of choice behind leaving education varied considerably. As we will show below, and as confirmed in the literature, long-term future plans are not always the most important factor in determining drop out (Johansson 2019).

Young people explained the vital conjuncture of leaving school through decision-making made in relation to their reading of their context, the possibilities—positive and negative—perceived within it, and their imagined futures. The reasons for leaving school may be grouped into four categories: financial problems; negative school experience (including performance-related); seeing alternative routes towards the future and employment (including understanding education as irrelevant); or due to interruption (such as marriage and displacement). In many narratives, these different reasons operated together.

For some research participants, the decision to leave school was definite and often related to experiences of failing in school, seeing themselves as incapable students, for financial reasons or, for some young women, because of marriage. However, as a vital conjuncture, leaving school was, for the majority of the research participants, not just a moment in time but a process that stretched out in time and less definitive. Leaving school

opened many alternative avenues while it closed others, and young people navigated these avenues in specific ways, as we show below.

The young people who shared their life histories with us showed that education is not a linear process. Even for those who thought they were leaving school definitely, many ended up continuing some form of education later on, emphasising that leaving education may be an interruption rather than a decisive moment in time. In our material young people predominantly value education, and even those who had not learnt to read and write and were now married and with children maintained a hope to go back to education. According to (Save The Children 2018), only 34% of children who leave school globally are likely to re-enrol in education, increasing to 38% in Arab countries. While this figure suggests that for many young people, education is interrupted rather than deserted, it also emphasises that only a minority of young people manage to return to school once they have left. Yet, far from being a sharp and clean-cut break with the education system, young people in our research demonstrate that leaving school is a process where several parallel (and sometimes crossing) paths in a young person's trajectory come together as young people move back and forth between education and employment or pursue both in parallel.

For many of the young people we interviewed, continuing or returning to education is one of their main aspirations. These dreams are maintained, often against the odds, and perceived as a way out of their current situation, or a bridge to an expected future. Within our sample, 57 of the interviewees (27 female, 30 male) discussed their attempts to return to education (by comparison, 117 interviews were coded as discussing not continuing education (dropping out or leaving), of whom 50 had not completed Grade 9/Grade 10). However, while many did manage to return to school, particularly if they had initially left while in primary or intermediate education (before Grade 9/Grade 10), their return was rarely a permanent or singular movement, with many leaving again within a year. This was often linked to the length of time spent out of education, and a perception that they would be too old or no longer had the energy for studying. The norms and morality of calendar time signify negative traits of 'delay', 'too old' and 'failure' and contribute to the difficulties in continuing education after interruption. Raed, an 18-year-old Palestinian man living in Saida, southern Lebanon, did not finish Grade 9. When asked why he did not return to school, he told us "I will not have the energy for that as before. If I had completed my studies and had still been learning, it would have been easy for me. But now, it is difficult for me because I quit school since a while".

Potential students, whose education had been interrupted for different reasons, were also put off by their reinsertion into the 'incorrect' year, either forcing them to repeat earlier grades among younger students or placing them in a grade that corresponded to their age, but not to their education thus far. Nasr, a 20-year-old Syrian man living in Amman (Jordan), told us:

> 'The principal asked me "to which grade I reached in Syria" and I said "Grade 5", and he asked "how many years did you stop school?" I said "2 3 years", so he said "at your age, you should be in Grade 8"'.
>
> **Would you have preferred to have stayed in Grade 5?**
>
> Yes, it would have been better for me.
>
> **Even if you were older than your classmates?**
>
> Even though! It wasn't a problem for me, I would have continued.
>
> **If you stayed in Grade 5?**
>
> Yes, I told him to put me in Grade 5 but he didn't accept'.

These tensions between lived time and bureaucratic calendar time are often aggravated by encounters with immigration systems (Hughes 2021). Similarly, in our material, young people were also affected by changing regulations regarding who could attend school,

re-starting education only to have it curtailed once again. Access to education was also limited by issues around presenting documentation.

For those who did return to school, teachers, parents and friends all played a key role in supporting their return, through helping with paperwork, explaining unfamiliar systems, assisting with homework and providing encouragement. In many cases, these 'cheerleaders' (Webb 2019) were seeking to support students to 'reinsert' themselves into a linear education pathway, to catch up on lost time. However, others provided support and validation for pursuing non-linear educational trajectories, and in doing so opened a possibility for young people to return or continue their education. Other young people wanted to continue their education through a homeschool modality, often due to negative earlier experiences in school, or gendered restrictions on their movement, but were put off by regulations (in Jordan) that require a student to be out of school for three years before they can enter homeschooling. For those from families with financial problems, money was a key determining factor in explaining why they did not return to education, even if the initial reason for leaving had a different origin. Relatedly, among the participants in our interviews, those who had fully stopped school to start working rarely returned to full-time education, instead often pursuing short vocational courses, emphasising the value of providing support for non-linear education pathways. Particularly for women, getting married also marked the end of the possibility to return to education, though a small number did continue their education after marriage and some continued to hope for a return to education.

Throughout their movement in and out of education, aspirations to return to school are often maintained, but seen as impossible or pursued alongside other potential pathways. While for some a different life event—such as marriage or starting work—may alter these aspirations, or re-calibrate their prioritisation, many continue to harbour a dream of continuing education, which is reflected in their aspirations for their children and the generational passing down of such dreams. Hence, the role of education in opening new possibilities and providing a 'way out' persists, with a sense of social responsibility and obligation also playing a role in the interest to return. Fadi, a 24-year-old Palestinian living in Lebanon, dropped out of university before deciding to return and complete his degree at a different institute.

> **'So, after working in painting, you had decided to go back to education?**
>
> Yes, sure. I decided to continue my education because I'm willing to have a degree even if I will not work with it.
>
> **Why?**
>
> It was mom's wish to educate me and I believed that I had to because I finished all of that and I should not quit. I wanted to satisfy my mom's wish. She wanted me to study and then to do whatever I want'.

Young people do not experience leaving education in a singular way, but the stretched out process of leaving can be used to unpack how to understand a vital conjuncture—these crucial moments in lived time where a trajectory may change direction. Trajectories of education in the context of displacement and crises are intricate and highly contextual. In the following, we aim to analyse in-depth two young people's narratives of education trajectories in order to demonstrate the ways in which different times come together in the process of leaving school as a vital conjuncture. As we showed above, in Lebanon, young men and those from refugee backgrounds are significantly more likely to leave school without gaining any qualifications (Shuayb et al. 2021). Similar effects were found in Jordan, albeit to a lesser degree. We have therefore chosen to analyse the narratives of two young displaced men—Salem and Ali, a Palestinian in Lebanon and a Syrian in Jordan—who, due to their displacement status, both have restricted access to the formal labour market. The two young men have distinctly different trajectories but share a prolonged and interrupted process of education.

Salem: Salem is a 17-year-old Syrian man living in Amman, Jordan. He left his home in Syria in 2011/2012 when he was 8 years old, shortly after completing the 2nd grade at school. After two years of moving between villages, he crossed into Jordan with his family, who were helped by extended family members living in Amman. During this time, Salem did not attend school. Once in Amman, he was placed into the afternoon shift in the 5th grade in a public school. Salem was bullied at school due to being Syrian, and after one year he was asked to leave the school following a fight with another student. While he managed to pass his Grade 5 exams before leaving, he lost the certificate and was unable to get a replacement. After a short time outside of school, his family moved to a new city, and Salem re-joined school. However, as he had lost his certificate, he had to repeat Grade 5. At the new school, Salem experienced even more extreme bullying than previously and, after a fight on the school bus, told his father he no longer wanted to continue school. He was also facing a move to the next stage of education, which would require a change to a larger single-sex school with a 'bad reputation' where he was concerned that bullying would be more prevalent. He told his father he wanted to learn a profession, and his father advised him to drop out.

Upon leaving school after Grade 5, he spent about a year at home, learning English and studying religion with his father. When the family returned to Amman, he first worked informally in the vegetable market, and then in different shops, including one run by a family member. Each of these positions was only for a few months at a time. After approximately four years out of education, working in informal short-term jobs, Salem began to regret leaving education. Around the same time, he fell in love with a distant relation he knew from childhood and wanted to ask her to marry him but felt he would not be accepted without educational qualifications. In 2018, he joined Questscope, an organisation that provides non-formal education through accelerated learning to prepare youths to get back into the formal school system. When we interviewed him, he had completed the initial months of Questscope's programming and was waiting to be admitted into a partner school providing non-formal education that would allow him to complete his Grade 10 equivalency in the coming years.

Ali: Ali is a 24-year-old Palestinian man living in a Palestinian refugee camp in southern Lebanon. His parents left education after Grade 6, and his siblings have reached grade 7, 8 and 9. Ali studied in an UNRWA school from Grades 1 to 6. Then, his school closed and he had to move to another school, where he studied in Grade 7. He failed the Grade 7 exams twice, after which the school let him pass to Grade 8. For Grade 8, he decided not to enrol, as he was unable to read or write and felt the school was not providing him with a quality or relevant education. His family was against him leaving school, particularly his maternal uncle, but they agreed once they understood that he was illiterate and that he intended to enrol in a vocational institute. After leaving school, Ali enrolled in a vocational institute to learn literacy. The course was for two years but after one year, at about 15 years old, he stopped. By then he could read and write in Arabic, but his English was weak. He took different jobs, such as a carpenter and working in a bakery, mainly inside the camp, finding work by going around asking for employment.

In 2011, when he was 17, his father went to work in Qatar and the family joined him. Without Ali's knowledge, his father had prepared all the documents from the Ministry of Education and Higher Education (MEHE) in Lebanon and the Qatari embassy for him to continue his studies in Qatar. They stayed in Qatar for seven years. Ali went to school and passed Grade 8 and 9 but failed Grade 10. He thought of re-doing Grade 10 as a home exam but did not and he also considered joining vocational training, but it was too costly in Qatar. Instead, he worked in a warehouse selling car products, a job found through a Palestinian connection. When Ali was 23, the family moved back to Lebanon. He said that by then he had passed the age limit for going back to school in Lebanon, and hence only vocational training was available to him. Initially, he wanted to do accounting, but people around him convinced him to pursue nursing. He said, '( . . . ) people made me love nursing. They told me it is a good major. So I loved it, and I started dreaming about it.

I started seeing myself as a nurse'. Partly due to the good job chances for Palestinians in nursing, because it is one of the occupations Palestinians can work in, and partly because of a scholarship that covers some of the training costs, nursing was his choice.

After much effort, he obtained a certificate that proved he had completed Grade 9 in Qatar, enabling him to complete the nursing degree over three years rather than five. At the time of the interview, he led a busy life. Alongside his nursing studies, he studies English three times a week, does an internship at two hospitals in the camp and is a volunteer in a youth organisation in the camp. The organisation provides a bursary of LBP 50,000 a month for the volunteering work, which is a passion he discovered when he was in Qatar, where it was an obligatory part of his education.

### 6. Leaving School as a Vital Conjuncture

As an analytical concept, 'vital conjunctures' can help to capture lived time and the interaction between being and becoming: it is a socially constructed zone of possibility that emerges around specific periods of potential transformation in life. We next interpret Salem and Ali's experiences in the light of this conceptual framework.

#### 6.1. Leaving School: Experiencing and Making Time

During the interviews, Salem and Ali reflected on their pasts and how they reached their current situation—how they had made and experienced their lived time thus far.

As we showed above, calendar time is the sequence of events within trajectories, the pauses, jumps and the speed with which young people move through events. Calendar time provides a publicly shared scaffolding that shapes—bureaucratically and socially—the possibilities that are present within conjunctures, and provides landmarks through which young people describe their current positions and orient their aspirations. In the two young men's narratives, the interruptions in education significantly affected their experience of time, which demonstrates the normative aspects of time: in the delays (losing time/value of time), their age being out of sync with expected educational or life positions, and becoming too old to take up education. In the process of leaving education as a vital conjuncture, leaving is not one moment in calendar time, but rather represents a stretched-out moment within which young people navigate. While in some cases the final decision may occur within one decisive moment, the origination of this decision and the navigation of the possibilities it opens are more extended.

Salem would have been in Grade 4 at the age when he entered school in Jordan, but instead was placed in Grade 5, without having undergone a placement test. His 'reinsertion' into education was done according to linear calendar time, based on his age and grade progression, but failed to take into account the gaps and 'lost years' in his education. When he left school, he had not progressed past Grade 5. His vital conjuncture of leaving school was completed—in his own experience—when he left for the third time. Contemplating his recent return to education, Salem continues to feel out of sync with calendar time. While Grade 10 exams are usually sat when students are about 16 years old, Salem plans to obtain his certificate by the time he is 19. This demonstrates a remarkable 'catching up' given the length of time he has been out of school, hence obscuring the biographical value of his time outside of school. It is as though all the calendar time in between was lost time, insignificant for the schooling system. Yet, this biographical (or narrative) time, which is currently not significant in the school system, continues to be meaningful for Salem: 'I'm 17 years old but I witnessed a lot in this life and learned a lot of lessons'.

Both Salem and Ali mention being 'too old' to go back to school. Salem felt that he 'will never be able to go back to school . . . I cannot go back to formal education, because I will go back to fifth grade', implying that returning to Grade 5 as a 16-year-old was socially unacceptable. Ali thought he had passed the age limit for going back to school in Lebanon and would have been refused enrolment due to bureaucratic restrictions at UNRWA schools. Ali re-entered school in Qatar at the same grade at which he had stopped in Lebanon, with the missed years and his older-than-average age for his school grade

not being mentioned as an issue, unlike in Lebanon. Calendar time is further evident in Ali's conjuncture in the repetition of grades, and in his expectation—shared with his parents—that a student should be able to read and write by the time they complete Grade 7.

The importance of the time of the trace, present through tangible proofs—certificates and documentation—of the time they have made was repeated in many of our interviews. Particularly in the context of displacement, some students were prevented from continuing school based on missing documentation or the inability to complete their exams before displacement. The importance of certificates is shown through Ali's tedious search for documentation of his education in Qatar and is related to his feeling of losing time. He emphasises that if he had not been able to obtain equivalency for his education certificates from Qatar, he would not have chosen nursing on his return to education in Lebanon, as he does not have time for the additional two years (five years instead of three) this would have required.

The experience of calendar time interacts with traces that shape the flow and direction of a trajectory. For example, the displacement events—from a distant or near past—are traces that stay with the two young men's trajectories through their different displacement and mobility histories. Their current marginal status due to displacement influences their access to education and employment, missing out on school, returning to school at an older age, losing certificates and employment options. As the young men grow older, they collect additional traces that they bring with them: experience of work-life, expanding social networks, widening horizons and changing aspirations for the future influence the way the young men value education.

Calendar time may be experienced as being forced into the norm of linear time—a feeling of losing time and making up for lost time is brought out by expectations regarding the timeliness of an achievement: of doing the right thing at the right time. These expectations from family and friends also represent generational time in young people's trajectories. For Salem, this is manifested firstly in his parents shifting attitudes towards schooling and employment, secondly in his intention to earn money and help the family, and thirdly in his plan to marry. His intention to marry is compromised by being late in education and not completing schooling, which would symbolise a crucial step towards adulthood, and independence from parents and the older generation. However, such expectations, where linked to the education system, also represent the shared calendar time that structures the expected progress and accumulation of 'meaningful' time through an engagement in education (Poole and Riggan 2020).

Vital conjunctures are created through the interplay of different times. Interpreting Salem and Ali's trajectories through lived time, we see that the continuous progress of calendar time often does not correlate with perceived movement within narrative time. Similarly, the expectations contained within generational time are often connected to the socially-shared calendar landmarks provided through calendar time (such as finishing a certain level of schooling before marriage), but also operate according to their own norms of 'the right time'. By evidencing the accumulation of other forms of time, the time of the trace maintains the past in the present. While these times may cohere, Ali and Salem's narratives reveal to us the more common sense of fracture and inconsistency between these senses of time. Prominent tensions between senses of time can lead to vital conjunctures being identified as breaking points.

### 6.2. Navigating in a Moving World: Zones of Possibility

The two young men are making time and moving towards adulthood, they aspire to use their education to do good and contribute to society. In their education decisions, the social value of education is highly significant. As mentioned above, aspirations for the future are not always the main determinant for leaving school. Many choices are highly contextual and made by navigating a changing present condition.

In both narratives, the role of their families as 'cheer leaders' (Webb 2019) and spokespersons at those crucial moments in time are essential in making their time in education, as Ali says about his re-entering school in Qatar:

'I didn't know anything. When we first arrived to Qatar, my dad woke me up one day in the early morning. I asked him why he is waking me up that early. He told me that I have to go to school. I told him what school? He told me that he has registered me in a school there. I asked him "really??", he said "yes". So I continued my studies in Grade 8 in Qatar. ( . . . ) I was very happy and surprised. I did not believe how all of it happened'.

For Salem, it is the immediate changes to his life world through bullying, harassment, the treatment by teachers and finally being asked to leave that are most salient. The zone of possibility that opens up for Salem when he leaves school for the third time is mainly a restricted job market amid limited life chances for refugees. Similarly for Ali, the decision to leave is based on an ambition 'to work and learn a profession', and schooling at that moment in time did not seem like a relevant activity for the present or the future. However, when Ali fails Grade 10 and leaves school in Qatar, it is experienced as a change of direction and as restricting his opportunities. In both young lives, the value of education is adjusted with the changing environment around them. For Ali, education is about learning, for its own sake and for functional skills, and the need for education to enable a contribution to society and consequently an investment in social standing; for Salem, education is not considered relevant for learning a profession, but over time becomes closely connected to the possibility of marriage and consequently also about social position. In this way, both young men use education strategically to manoeuvre the changing landscape they experience as they grow older and their orientations change.

Possibilities and aspirations beyond education are also changing during the stretched-out process of leaving. The options identified during this vital conjuncture contribute to a shift in aspirations. Included in the changing aspirations are the regrets of leaving school for both of them. Their differing experiences after leaving school in Qatar and Jordan increased their appreciation of education. The dissatisfaction with frequent changes in low-paid employment, seeing how peers can progress with education and realising the social standing that comes with education means that the vital conjuncture of leaving education shapes their possible pathways, including the options to return to education when they decide to do so.

Salem was very clear in interpreting his past: 'When I dropped out, at that time I did not think that I would deprive myself from education all my life, and I was young and I did not think thoroughly about the issue, I was just thinking of getting rid of the bad situation I was in. I really regretted leaving school'. A few months before we met, Salem had finally returned to education, acting against his family's advice, though with their support. He says that he has learned many lessons and his personality has changed, altering his decision-making, aspirations and self-image, and thereby the possibilities perceived within future conjunctures.

Similarly, Ali's dropping out of school in Qatar shaped his determination to continue his formal education. Having realised, partly through the experience of moving to Qatar, the value of education and the possibilities it might open up, he carefully planned how to get back to it. Ali evaluated the routes that held the greatest potential for his employment—and thereby the means to his other aspirations—with his age and legal status substantially shaping the possibilities he perceived. When he returned to Lebanon, it was the opportunities as a Palestinian that shaped his decision to become a nurse.

Many decisions are taken based on the immediate context, but navigating the zones of possibility is influenced by expectations regarding the future. When Ali returns to education, he is determined to get a diploma and to learn English, as he sees it as fundamental to get a job or go abroad to continue studies or work. While his plans are uncertain and contain multiple forms of education, including a simultaneous engagement in vocational training, language education and volunteering, they share a common direction and corre-

spond to the possibilities he sees for himself based on his context and in the constraints represented by the traces of his status as a Palestinian. As noted previously, despite Ali's determination and careful strategising, the context of Palestinian displacement in Lebanon suggests that he will face further challenges in pursuing these goals. Indeed, the multiple avenues he is pursuing—formal education, skill development, volunteering—reflect a strategy seen across many of the interviews, in which young people endeavor to create and maintain multiple options in the present and for the future, in the face of their uncertainty.

The two narratives we have analysed represent the embodiment of time, the changing conditions for young people generally, and specifically for those displaced: it is change as generational time. The value of education has increased between generations, sometimes instigated by the older generation, yet at other times shaped by the changing landscapes in which families and individuals move. Salem's story reveals shifting attitudes towards education within the relatively short duration of the vital conjuncture. Before his displacement, upon the family's entry into Jordan, and following his initial attempt at Grade 5, his parents encouraged and supported his education. It is when his father agrees with his desire to leave education that Salem does not attempt to continue in school. When he decides to return to education, it is his mother that tells him about the possibility of informal education, yet she also advises him that education is not right for him.

Experiences of lived time form the zones of possibilities open to Ali and Salem. The lost time, the missing certificates and the ever-present refugee status contribute to their paths towards informal education and nursing. Amidst changing possibilities and clearer aspirations as the two young men grow older, they realise the meaning of formal qualifications for marriage and social standing as much as for future work. For both of them, the making up of lost time through recovering certificates and catching up with schooling become central strategies for expanding the zones of possibilities.

If we consider all the 293 life histories shared with us, Salem and Ali belong to the most likely group in our material to drop out of education before gaining a certificate: They are also more likely to be unemployed or underemployed. At the time of our interviews, Salem and Ali confidently described their future plans. Their plans played an important role in the education paths they have chosen and that were open to them. Yet, these plans are imbued with uncertainty, as the young men continue to navigate changing legal, economic, political and social realities. As such, their trajectories are 'incomplete' and still in the making.

## 7. Studying Young People's Multidirectional Trajectories: Towards a Conclusion

In this paper, we have conceptualised how trajectories can be understood as 'lived time'. By analysing young people's trajectories through four types of time—narrative, calendar, time of the trace and generational—we are able to build an image of the multiple and multi-scalar influences on the vital conjunctures that plot their trajectory and how young people navigate the conjunctures. By integrating a multi-faceted understanding of temporality into our notion of trajectories, we have attempted to explore in more detail how a trajectory can be multidirectional and complex. The young men's narratives highlight how a trajectory is plotted by the interaction of different vital conjunctures, in this case particularly between their education and migration pathways. Rather than static designations of refugee status, the multiplicity of a 'lived time' understanding of trajectory allows us to see how young refugees navigate fluctuating environments and the constraints associated with their various positions, not only their legal status. The work then also emphasises a much under-researched dimension in the experience of displacement: time and temporality (Brun and Thorshaug 2020; Thorshaug and Brun 2019).

Analysing young people's education trajectories as a series of interconnected conjunctures emerging within different temporalities allows us to see that a trajectory is not just a calendar timeline and that a vital conjuncture has many internal and parallel paths that together create the new openings. Using lived time shows us that a vital conjuncture is not one moment in time (according to calendar time) but rather represents a stretched-out

moment. While narratives can synthesise and disguise the restlessness, indeterminacy and contingency of the past, understanding trajectories as lived time reveals how young people piece together their trajectories and the continuity and change within them. The value of the notion of lived time lies in opening up conjunctures to understand how different temporal elements come into play to produce these critical moments, how they interact within specific contexts to produce possibilities for young people, and in understanding how young people make decisions in relation to who they are and who they want to be.

Lived time is rarely considered in understandings of how and why young people leave and re-enter school. Young people who leave education are often viewed as 'blocked' or as having 'dropped out'. Here, we show that many young people view leaving education as an interruption, or engage in education and employment simultaneously, rather than as a decisive break. For others, leaving education serves as a way to focus on the immediate present and avoid the expected failure that awaits them. Young people may leave school with limited or no clear plans for what comes next, and leaving education is only marginally done with an eye to a long-term future. Rather, in leaving, as we have shown here, youths manoeuvre away from predicted difficulties at school and create spaces for new, alternative possibilities, albeit these are typically rather limited spaces. In understanding conjunctures as multi-temporal zones of possibility, we can recognise how young people make meaning in their present time, as well as how considerations of the past and future factor into these decisions. Given the worrying statistics of educational attainment and employment in the region, insights into how school leaving is practised and experienced should be taken into consideration when support is offered and policies are developed.

Our research points to a more nuanced understanding of the process of leaving school. It shows that much more can be done to accommodate education that is less disciplined by calendar time and that accounts for the biographical value of the interruptions that take place in the context of displacement and uncertainty. Analysing leaving school as a vital conjuncture, and within the transition from education to employment, our framework is also valuable for a more general understanding of the critical moments and transitions that constitute a young persons' trajectory As we have shown here, linear calendar time is but one aspect of the complex temporality of a life history which discloses the agency and navigation that takes place in the zones of possibilities open to young people.

**Author Contributions:** The two authors contributed equally to this paper. All authors have read and agreed to the published version of the manuscript.

**Funding:** ESRC/GCRF (project ref ES/S004742/1) and the IDRC (project ref 109043-001), is co-led by the Centre for Lebanese Studies (CLS) at the Lebanese American University, CLS Jordan and the Centre for Development and Emergency Practice (CENDEP) at Oxford Brookes University. The APC for this Special Issue was waived.

**Institutional Review Board Statement:** The study was conducted according to the guidelines of the Declaration of Helsinki, and approved by the Institutional Review Board of the Lebanese American University (IRB#: LAU.STF.MS1.3/Jan/2019 accessed on 4 January 2019).

**Informed Consent Statement:** Informed consent was obtained from all subjects involved in the study.

**Data Availability Statement:** Anonymised data will be available through a project radar held at Oxford Brookes University on the completion of the research project (October 2021).

**Acknowledgments:** We would like to thank the project team for their collective work in the design and implementation of this research, and for their review of earlier versions of this article. In alphabetical order: Hala Abou Zaki, Dina Batshon, Oroub El Abed, Nadim Haidar, Alexandra Kassir, Cyrine Saab, Hamza Saleh, and Maha Shuayb. We thank Johanna L. Waters for the thoughtful and inspiring comments she provided from the start of our project. We also thank the editors, Georgia Dona and Angela Veale for bringing this special issue together, and the valuable feedback they provided on earlier drafts. Most importantly, we thank all our research participants for their generosity in sharing their narratives and reflections with us.

**Conflicts of Interest:** The authors declare no conflict of interest. The funders had no role in the design of the study; in the collection, analyses, or interpretation of data; in the writing of the manuscript, or in the decision to publish the results.

## Notes

1   'Nationals' is understood here as those with a national ID card and number. This includes those of Palestinian origin who hold these documents. However, recognising that nationals with Palestinian origin may have a substantially different experience than nationals without Palestinain origins, the interviews included a discussion of family background and origin and perceived impacts of this on young people's trajectories.

2   Participants for interview were selected and contacted separately from those included in the survey.

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
