# Peer review of "Vital Conjunctures in Compound Crises: Conceptualising Young People’s Education Trajectories in Protracted Displacement in Jordan and Lebanon"

_socsci, doi:10.3390/socsci10070241_

Round 1
Reviewer 1 Report
Overall, I love this paper and think it is novel, fascinating, and makes a great contribution. This journal will be lucky to have it. The following are small points for improvement, not critiques:
1) "Vital conjuncture":
This seems very related to the comparative historical social science literature on critical junctures. If you are trying to reach political scientists with this article, consider adding in citations from that literature.
2) Lines 262-265:
This sentence is very long and has some awkward grammar - could rewrite and split in two
3) Line 266 or thereabouts:
What was interview language and how did you handle multi-lingual/translation contexts? Say more about language issues.
4) Around 267-that parargraph:
In the spirit of the reflexive turn in political science, I'd like to know more about the positionality of the interviewers. Did they speak same language/relate identity-wise to interviewees? Reflection on those issues would be helpful given how intimate a life history collection can be. See MacLean or Thomson from QTD conversation in political science to learn more about this turn.
5) P. 9-10:
This is all very interesting, but is it worth mentioning why you chose these two narratives out of 293? Are they representative? Did you cherry-pick? Can you justify the selection of these two in some way?
6) line 565 grammar issue?
Author Response
please authors responses in the attached file.

Reviewer 2 Report
This is very important and well written paper. My only remark is that the methods of conduct are not explained: What kinds of qualitative interviews have been conducted? Who conducted them? Where?
The methodology and the theoretical frame are plausible and the selected cases are impressive. It would be sufficient from my perspective to add a paragraph on matters of conduction and interview-techniques as well as sampling.
Author Response
please find author responses in the attached file.

Round 2
Reviewer 1 Report
I appreciate the thoughtful ways in which the authors addressed my previous comments and the draft is much improved. I recommend for publication and don't think it requires more than a light copy edit to move ahead. However, I leave the authors with two comments, neither of which have to be addressed for publication, but I offer them in the spirit of intellectual engagement.
1) I take the point that critical junctures are usually institutional and they posit vital conjunctures as something more individual. But I do think homage should be paid to a term that is remarkably similar, and ultimately is talking about the same thing - a fork in the road that is deterministic. Adding in a Mahoney citation or someone else doesn't seem to overcomplicate in my view.
2) I see the authors added one line about the diverse identities of RAs, but this is a relatively superficial treatment of positionality in light of a new and significant turn towards researchers situating their own identity in research. I personally find their addition insufficient and would recommend adding several more sentences about positionality and power dynamics in relation to the researchers. See: MacLean, Lauren M., Posner, Elliot, Thomson, Susan, & Wood, Elisabeth Jean. (2018). Research Ethics and Human Subjects: A Reflexive Openness Approach. Retrieved from Available at: https://papers.ssrn.com/sol3/papers.cfm?abstract_id=3332887:
Over all, it is an important article and the journal is lucky to have it.
This manuscript is a resubmission of an earlier submission. The following is a list of the peer review reports and author responses from that submission.
Round 1
Reviewer 1 Report
Overall, I love this paper and think it is novel, fascinating, and makes a great contribution. This journal will be lucky to have it. The following are small points for improvement, not critiques:
1) "Vital conjuncture":
This seems very related to the comparative historical social science literature on critical junctures. If you are trying to reach political scientists with this article, consider adding in citations from that literature.
2) Lines 262-265:
This sentence is very long and has some awkward grammar - could rewrite and split in two
3) Line 266 or thereabouts:
What was interview language and how did you handle multi-lingual/translation contexts? Say more about language issues.
4) Around 267-that parargraph:
In the spirit of the reflexive turn in political science, I'd like to know more about the positionality of the interviewers. Did they speak same language/relate identity-wise to interviewees? Reflection on those issues would be helpful given how intimate a life history collection can be. See MacLean or Thomson from QTD conversation in political science to learn more about this turn.
5) P. 9-10:
This is all very interesting, but is it worth mentioning why you chose these two narratives out of 293? Are they representative? Did you cherry-pick? Can you justify the selection of these two in some way?
6) line 565 grammar issue?
Author Response
Thank your for your helpful review and the positive reflections on our paper. Please see below for a table summarising our response to your suggestions.
|
Line |
Comment |
Author Response |
|
1) "Vital conjuncture":
|
This seems very related to the comparative historical social science literature on critical junctures. If you are trying to reach political scientists with this article, consider adding in citations from that literature. |
We have considered this comment and carefully gone through some of the literature, where similar intricate multidirectional processes may be found. However, the critical junctures literature is largely institutional and can easily be confused with the emphasis of understanding individual narratives in our work. Hence, we decided against including references to this work, as we felt it was complicating. |
|
2) Lines 262-265:
|
This sentence is very long and has some awkward grammar - could rewrite and split in two |
Sentence rephrased and split in two |
|
3) Line 266 or thereabouts:
|
What was interview language and how did you handle multi-lingual/translation contexts? Say more about language issues. |
Added note of interviews in Arabic, bi-lingual researchers, quality check of translation and transcription by original interviewers. |
|
4) Around 267-that paragraph:
|
In the spirit of the reflexive turn in political science, I'd like to know more about the positionality of the interviewers. Did they speak same language/relate identity-wise to interviewees? Reflection on those issues would be helpful given how intimate a life history collection can be. See MacLean or Thomson from QTD conversation in political science to learn more about this turn. |
Added some reflections on the wide range of positions within the research team to the text |
|
5) P. 9-10:
|
This is all very interesting, but is it worth mentioning why you chose these two narratives out of 293? Are they representative? Did you cherry-pick? Can you justify the selection of these two in some way? |
The choice is a reflection of the higher representation of refugee men in young people who left school before finishing basic education + representing Syrians and Palestinians, and Jordan and Lebanon. This has been added to the text - Lines 355 - 362 |
|
6) line 565 |
grammar issue? |
Corrected |
Reviewer 2 Report
Few problems:
- info: West Bankers/Gazans are accepted in public universities but as internationals (pay more).
- Theoretical part too long and abstract.
- Section 4: Nationals in Jordan include of Palestinian origin? Makes all the difference
- Too much disparity between sections on statuses (quite comprehensive) and the two case studies: why such a choice -except convenience.
- Section 5, line 81: when was the survey conducted? Nationality of respondents?
- Should emphasize more role of networks (and related time dimension).
Author Response
Thank you for the consideration given to our paper and your helpful suggested revisions. Please see below for a table summarising our response.
|
Line |
· Comment |
Author Response |
|
|
· info: West Bankers/Gazans are accepted in public universities but as internationals (pay more). |
Added clarification as footnote |
|
|
· Theoretical part too long and abstract. |
We have considered this comment. However, we had already stripped the words considerably before submission. Also because we consider (as do other reviewers) this one of our main contributions, we would like to keep the section largely unchanged. The section presents how we have developed a theoretical framework and whilst abstract, the different times and concepts have been exemplified. |
|
Section 4: |
· Nationals in Jordan include of Palestinian origin? Makes all the difference |
Yes, nationals includes those of Palestinian-origin. This distinction was discussed with participants during interviews and is noted in our database. This clarification has been added to the text. |
|
|
· Too much disparity between sections on statuses (quite comprehensive) and the two case studies: why such a choice -except convenience. |
Details added – lines 335 - 362 |
|
Section 5, line 81: |
when was the survey conducted? Nationality of respondents? |
Details added to text (line 298) |
|
|
Should emphasize more role of networks (and related time dimension |
In the analysis of the two narratives, we have included networks and social connections where they need to mentioned and in relation to the related time dimension. We do not find it appropriate to emphasise this more given the aim of the paper. |
Reviewer 3 Report
This is very important and well written paper. My only remark is that the methods of conduct are not explained: What kinds of qualitative interviews have been conducted? Who conducted them? Where?
The methodology and the theoretical frame are plausible and the selected cases are impressive. It would be sufficient from my perspective to add a paragraph on matters of conduction and interview-techniques as well as sampling.
Author Response
Thank you for your review of our paper and the helpful suggestions for improvement. In line with this, we have added information on the type of interviews conducted (line 254), who conducted them and where (line 267 – 269), and sampling (273 to 275).